# Clinical and Microbiological Characteristics of Neonates with Candidemia and Impacts of Therapeutic Strategies on the Outcomes

**DOI:** 10.3390/jof8050465

**Published:** 2022-04-29

**Authors:** Yu-Ning Chen, Jen-Fu Hsu, Shih-Ming Chu, Mei-Yin Lai, Chih Lin, Hsuan-Rong Huang, Peng-Hong Yang, Ming-Chou Chiang, Ming-Horng Tsai

**Affiliations:** 1Division of Pediatric Neonatology, Department of Pediatrics, Chang Gung Memorial Hospital, Taoyuan 333, Taiwan; nono2261@cgmh.org.tw (Y.-N.C.); jeff0724@gmail.com (J.-F.H.); kz6479@cgmh.org.tw (S.-M.C.); lmi818@msn.com (M.-Y.L.); qbonbon@cgmh.org.tw (H.-R.H.); cmc123@cgmh.org.tw (M.-C.C.); 2College of Medicine, Chang Gung University, Taoyuan 333, Taiwan; annielin85@gmail.com (C.L.); ph6619@cgmh.org.tw (P.-H.Y.); 3Department of Laboratory Medicine, Chang Gung Memorial Hospital at Linkou, Taoyuan 333, Taiwan; 4Division of Neonatology, Department of Pediatrics, Chang Gung Memorial Hospital, Keelung 204, Taiwan; 5Department of Pediatrics, Chang Gung Memorial Hospital, Chiayi 613, Taiwan; 6Division of Neonatology and Pediatric Hematology/Oncology, Department of Pediatrics, Chang Gung Memorial Hospital, Yunlin 638, Taiwan

**Keywords:** candidemia, intensive care unit, antifungal resistance, bloodstream infection, invasive candidiasis

## Abstract

Neonatal candidemia is associated with significant morbidities and a high mortality rate. We aimed to investigate the clinical characteristics of *Candida* bloodstream infections in neonates and the impact of therapeutic strategies on the outcomes. We identified all the neonates with candidemia from a medical center in Taiwan over an 18-year period (2003–2021) and analyzed them. Clinical isolates were confirmed by DNA sequencing, and antifungal susceptibility testing was performed. The prognostic factors associated with clinical treatment failure (30-day, all-cause mortality and persistent candidemia > 72 h after antifungal agents) and in-hospital mortality were analyzed using logistic regression modeling. A total of 123 neonates with 139 episodes of candidemia were included in the study. The median (IQR) gestational age and birth weight of the neonates with candidemia were 29.0 (26.0–35.0) weeks and 1104.0 (762.0–2055) g, respectively. The most common *Candida* spp. was *Candida albicans* (*n* = 57, 41.0%), followed by *C. parapsilosis* (*n* = 44, 31.7%), *Candida guilliermondii* (*n* = 12, 8.6%), and *C. glabrata* (*n* = 11, 7.9%). The overall susceptibility to fluconazole was 81.3%, and the resistant rates against other antifungal agents were less than 3%. The cumulative mortality rate at 7 and 30 days after the first episode of candidemia was 11.3% and 32.3%, respectively. The overall in-hospital mortality rate was 42.3%. The treatment outcomes did not change over the study period and were not affected by delayed initiation of antifungal agents. Multivariate analysis showed that delayed catheter removal (odds ratio [OR], 5.54; 95% confidence interval [CI]: 1.93–15.86, *p =* 0.001), septic shock (OR, 7.88; 95% CI: 2.83–21.93, *p* < 0.001), and multiple chronic comorbidities (OR, 8.71; 95% CI: 1.82–41.81, *p =* 0.007) were independently associated with the final in-hospital mortality. We concluded that the overall mortality of neonatal candidemia has remained consistently high over the past decade. Prompt early catheter removal and an aggressive treatment strategy for neonatal candidemia with septic shock would be critical to improving patient outcomes.

## 1. Introduction

Bloodstream infections (BSIs) caused by candida species remain a substantial burden in the hospital and are associated with high morbidity and mortality rates among intensive care unit (ICU) patients [1,2]. Invasive candidiasis or candidemia deserves greater concern because it is associated with a high mortality rate of 35–60%, especially in critically ill neonates [3,4,5]. Recent population-based surveillance studies have shown an increasing incidence of invasive candidiasis in the neonatal ICU during the past decade [6,7]. An increased survival rate of extremely preterm neonates with long-term hospitalization and the widespread use of broad-spectrum antibiotics may account for the increasing incidence of invasive candidiasis [8,9,10]. Furthermore, more antifungal prophylaxis and the empiric use of echinocandins are reported to be associated with a continuous shift from *C. albicans* to various non-*albicans Candida* candidemia, which are more likely to be azole-resistant [11,12].

There have been numerous studies that describe the epidemiology, clinical features, antifungal treatment, and outcomes of neonates with candidemia [6,7,8,9,10]. However, few researchers have investigated the reasons for the treatment failure of neonatal candidemia [5,12,13,14]. The effects of the different therapeutic strategies on the final outcomes of candidemia are often concluded from adult patients or studies conducted in the surgical and pediatric ICUs or the hematological wards [2,15,16]. Additionally, the studies of neonatal candidemia were often limited by small sample sizes or lack of antifungal resistance or did not analyze the impact of different therapeutic strategies on outcome [2,6,7,8,9,10,13,14,15,16]. The aims of this study were to describe the clinical characteristics of neonates with candidemia, to present the results of antifungal susceptibility testing, and to assess the influences of different therapeutic strategies on the outcomes.

## 2. Patients and Methods

### 2.1. Study Population, Setting, and Design

All the hospitalized neonates admitted to the NICU of Linkou Chang Gung Memorial Hospital (CGMH) from January 2003 through December 2020 with documented candidemia were enrolled for analysis. Candidemia was defined as being present in those with ≥1 blood culture positive for *Candida* species and who had symptoms and signs compatible with candidemia. All patient demographics, hospital courses, predisposing factors within the preceding 30 days from the onset of candidemia (defined as the day of the first positive blood culture for *Candida* species), clinical managements, and outcomes were retrospectively investigated and recorded. The study was approved by the Institutional Review Board and Human Research Ethics Committee of CGMH, and a waiver of informed consent for anonymous data collection was approved.

### 2.2. Definitions

An incident episode of candidemia was defined as a ≥1 positive blood culture drawn from a peripheral vein yielding a *Candida* isolate, with clinical symptoms and signs compatible with *Candida* BSI [2,7,15]. Episodes were considered to be separate if they occurred ≥ 1 month apart [17,18]. Catheter-related *Candida* BSI was diagnosed when the catheter tip culture was positive for the same *Candida* species as those obtained from the peripheral vein and there was no evidence of infection at another site [19]. Persistent candidemia was defined as a positive blood culture of candida species for > 5 days [20]. The severity of illness was measured by the Neonatal Therapeutic Intervention Scoring System (NTISS) score on the day of candidemia [21], and the presence of severe sepsis or septic shock at presentation was defined based on the standard criteria [22].

The primary study outcome was clinical treatment failure, which was defined according to the Mycoses Study Group and European Organization for Research and Treatment of Cancer consensus criteria [23], as the following: (1) all-cause mortality between days 3 and 30 from the initial positive blood culture or (2) persistent fungal BSI for ≥72 h after the initiation of antifungal therapy. Patients who died within the first 72 h were excluded from the analysis of outcome predictors to ensure that the potential impact of therapeutic strategies could be appropriately investigated. The secondary outcome of in-hospital, all-cause mortality was also analyzed.

### 2.3. Microbiological Studies

All *Candida* isolates were processed to have species re-identification using Matrix-assisted laser desorption ionization time-of-flight mass spectrometry (MALDI-TOF, Bruker Biotype, software version 3.0, San Jose, CA, USA) and molecular methodology by sequencing the internal transcribed spacer regions (ITS1 and ITS2) from ribosomal DNA. Therefore, the identities of *Candida parapsilosis* sensu stricto, *Candida orthopsilosis*, and *Candida metapsilosis* isolates were confirmed. The in vitro antifungal susceptibilities of the isolates were evaluated according to the EUCAST-Antifungal Susceptibility Testing microdilution method [24,25]. *Candida krusei** ATCC^®®^ 6258 and *Candida parapsilosis* ATCC^®®^ 22,019 were used as the quality control strains for the antifungal drug susceptibility testing.

## 3. Statistical Analysis

The χ^2^ test or the Fisher exact test were used for the categorical variables, and the student *t* test or Mann–Whitney U test were used for the continuous variables. All the significant tests were 2-tailed, with a *p* value of less than 0.05 to be significant. We analyzed the impact of the initial treatment strategy on the primary outcome, with the predictors for clinical treatment failure assessed in the entire study population by using a backward stepwise logistic regression model.

For the secondary outcome, the follow-up period was until death or discharge from hospital to evaluate the variables related to death. A univariate logistic regression was fitted for each variable to test its relationship with the mortality outcomes. Variables that were clinically relevant and statistically significant (*p* < 0.1) on univariate analysis were enrolled into the multivariate regression model. Clinical interventions were maintained in the final model as a fixed variable. Potential confounders of the treatment strategies (NTISS score) were tested. We also ruled out significant interactions between the variables. Statistical analyses were performed using the SPSS, version 21.0 (IBM SPSS, Chicago, IL, USA).

## 4. Results

A total of 148 episodes of candidemia were identified during the study period. Among them, nine patients were excluded because the *Candida* species were unidentified in four cases and there were missing data from the hospital courses and/or final outcomes in five cases. Therefore, we enrolled a total of 139 episodes of candidemia in 123 neonates hospitalized in the NICUs of CGMH for analyses. The median (interquartile range, IQR) gestational age and birth weight of the neonates with candidemia were 29.0 (26.0–35.0) weeks and 1104.0 (762.0–2055) g, respectively. The median onset of candidemia in our cohort was 29.0 days old (IQR, 15.0–53.0 days), and there were seven episodes of early-onset sepsis (within the first week of life). Most of the patients had underlying chronic comorbidities, and more than one-third (*n* = 47, 37.9%) had more than one comorbidity. The patient demographics of the study subjects are summarized in Table 1.

### 4.1. Microbiological Characteristics and Antifungal Susceptibility Results

Among all these candidemia episodes, there were two episodes in which two different *Candida* species were identified simultaneously on the same fungal culture. Mixed candidemia/bacteremia episodes were noted in 14 cases (10.1%), and Gram-negative rods were the most common bacterial pathogens (*n* = 9), followed by coagulase-negative staphylococci (*n* = 4) and other Gram-positive cocci (*n* = 1). Most of these episodes were primary bloodstream infections (*n* = 92, 66.2%) (Table 1). Forty-six point eight percent (*n* = 65) had other infectious focuses that led to secondary candidemia, including catheter BSIs (*n* = 37), intra-abdominal (*n* = 14), pleural fluid (*n* = 7), urinary source (*n* = 4), and meningitis (*n* = 3). There was a total of 12 episodes of disseminated candidemia, which indicated that the *Candida* isolates were identified from more than two sterile sites. During the study period, there was an increasing trend of non-*albicans* candidemia, and this trend was statistically significant (*p* < 0.01) (Figure 1).

The antifungal susceptibility results of all *Candida* isolates that caused neonatal candidemia during the study period are presented in Table 2. Overall, 81.3% of all *Candida* isolates (*n* = 113) were susceptible to fluconazole (minimum inhibitory concentration [MIC] < 4 mg/L), although some uncommon Candida species have no standard MIC cut-off point of resistance. Specifically, two (3.5%) *C. albicans* isolates were resistant to fluconazole, but 100 % of *C. glabrata* and 40.0% of *C. tropicalis* were intermediate or resistant to fluconazole. As we suspected, non-*albicans* Candida isolates had a significantly higher rate of azole resistance (*p* < 0.01). All Candida isolates were susceptible to amphotericin B, and resistance to anidulafungin, caspofungin, and micarfungin was uncommon: 2.3% for *C. parapsilosis* (*n* = 1), 9.1% for *C. glabrata* (*n* = 1), and no resistance among *C. albican* and *C. tropicalis*. However, the MIC_90_ of echinocandins against *C. parapsilosis* and *C. glabrata* was higher than those recorded for the most common Candida species. Because of the increasing trend of non-*albicans* candidemia during the study period and a significantly higher rate of antifungal resistance among non-*albicans* candida species, there was also an increasing trend of azole-resistant candidemia during the study period. 

### 4.2. Clinical Manifestations and Treatments

Among these candidemia episodes, 77 (55.4%) and 55 (39.6%) presented with severe sepsis and septic shock, respectively. After effective antifungal agents, 41 (29.5%) had a progressively deteriorated course, and 14 cases had persistent candidemia, which had positive *Candida* strains recovered from more than two consecutive fungal cultures after effective treatments. Four had an obstructing renal fungus ball and one had septic thrombophlebitis during the follow-up period. Blood cultures were persistently positive for more than 3 days in 37.4% of all the candidemia episodes.

The standard dosing of antifungal agents was prescribed in 134 (96.4%) of all the neonatal candidemia episodes, and there were five patients who did not receive any antifungal agents because all these patients died before the blood cultures of the *Candida* isolates were informed (Table 3). Fourteen point two percent were breakthrough candidemia; these patients were on antifungal prophylaxis when candidemia occurred. Antifungal therapy was initiated after a median of two days (range, 0–7), following the drawing of the first diagnostic blood culture. Of the 134 episodes with antifungal treatment, 39.6% (*n* = 55) had modification of the antifungal therapeutic agents, depending on the physicians’ decisions. The median duration of all therapeutic courses was 17.0 days (IQR, 14–24 days and range, 1–68).

After the documentation of candidemia, the CVC was removed within 48 h and 72 h after obtaining the first positive blood cultures in 23.7% (33 of 139) and 36.0% (50 of 139) of the episodes, respectively. Neonatal candidemia with severe sepsis and septic shock was more likely to have delayed CVC removal (CVC removal performed after 72 h of onset) (80.5% vs. 43.6% and 80.0% vs. 53.6%, *p* < 0.001 and *p* = 0.002, respectively), which may be due to critically ill situations. After excluding five cases without any antifungal treatments, 33.1% (*n* = 46) of the episodes of candidemia were treated with a catheter in situ, and the last 27.3% (*n* = 38) of the episodes had elective CVC removal. Among those with elective CVC removal, 21 (55.3%) of them had symptoms of candidemia, and clearance was resolved only after CVC removal.

### 4.3. Therapeutic Responses and Predictors of In-Hospital Mortality

Cumulative mortality at 7 and 30 days after the onset of candidemia was 11.4% (*n* = 14) and 32.3% (*n* = 40), respectively. Overall, the candidemia-attributable mortality rate was 24.2% (*n* = 30) and the in-hospital mortality rate for the neonates with candidemia was 42.7% (*n* = 52). The Kaplan–Meier method was used to compare the attributable mortality rates of sepsis caused by different pathogens in the NICU (Figure 2). Neonatal sepsis caused by Candida species had a significantly higher risk of overall mortality compared with other microorganisms (*p* < 0.01 by log rank test) (Figure 2). Although new antifungal agents were introduced in 2012, most therapeutic strategies have not changed in the past two decades, and the therapeutic outcomes were comparable between the four study periods (2003–2006, 2007–2011, 2012–2015, and 2016–2020).

Candidemia caused by non-albicans *Candida* species was previous reported to be associated with higher rates of antifungal resistance and treatment failure, but the differences of treatment were not observed in our cohort (Table 3). We identified the risk factors of treatment failure for neonatal candidemia using multivariate logistic regression (Appendix A). We found that delayed CVC removal (odds ratio [OR], 4.14; 95% confidence interval [CI]: 1.52–11.23, *p* = 0.005) and breakthrough candidemia (OR, 4.97; 95% CI: 1.80–13.7, *p* = 0.002) were independently associated with clinical treatment failure.

The independent risk factors of final in-hospital mortality in the neonates with candidemia were examined after excluding those without antifungal treatment. Gestational age, final antifungal treatment, and candida species were not independently associated with the final outcomes (Table 4). After multivariate analysis, the independent risk factors for in-hospital mortality in the neonates with candidemia were delayed catheter removal (odds ratio [OR], 5.54; 95% confidence interval [CI]: 1.93–15.86, *p* = 0.001), septic shock at the onset of candidemia (OR, 7.88; 95% CI: 2.83–21.93, *p* < 0.001), and multiple chronic comorbidities (OR, 8.71; 95% CI: 1.82–41.81, *p* = 0.007).

## 5. Discussion

Most of the current guidelines and the impacts of therapeutic strategies on the outcomes of invasive fungal infections are concluded from adult studies, patients with hematological malignancy, or surgical intensive care units [26,27]. There are relatively few data regarding neonatal candidemia, and the influences of illness severity, catheter removal, and the timing of antifungal therapy have not been well investigated [9,26,27]. We found that the treatment failure rate and final mortality rate among neonates remain high, especially in cases with underlying chronic comorbidities and catheter in situ. After multivariate analysis, we found delayed catheter removal, multiple chronic comorbidities, and septic shock at the onset were independently associated with final mortality among neonates with candidemia.

The in-hospital mortality and candidemia-attributable mortality rates of our cohort were relatively higher than previous studies, most of which reported a mortality rate of between 11.4% and 44%. The higher mortality rate in our cohort could be explained by the higher rates of severe sepsis and septic shock in our cohort. We have documented that the neonates with candidemia had a significantly higher rate of septic shock at the onset than late-onset sepsis caused by Gram-positive or even some Gram-negative pathogens [28]. Additionally, most of the neonates with candidemia had underlying chronic comorbidities, which has been documented to be the independent risk factor of the final adverse outcomes [28,29,30].

The issue of whether the CVC should be removed as early as possible after the occurrence of sepsis has been the topic of debate as most critically ill patients need the CVC for antibiotic administration and fluid supplementation. Although there have been multicenter studies that did not reach a conclusion on the importance of early CVC removal to the outcomes [31,32,33], we did find that delayed CVC removal in neonates with candidemia was independently associated with final in-hospital mortality. This can probably be explained by these patients being more likely to have persistent candidemia and/or breakthrough candidemia, which is associated with final adverse outcomes [34,35]. However, in cases when the catheter is not the source of sepsis, it deserves a prospective, randomized controlled trial to document whether prompt catheter removal contributes significantly to the outcomes as it is often difficult to reinsert the CVC during the critically ill period. Because of different study designs, cohort selections, and perhaps the bias of illness severity, previous studies failed to have undebatable conclusions [31,32,33,36]. Nevertheless, we suggest that CVC removal at earliest stage of sepsis should be considered for neonates with candidemia, based on our study result and the current guidelines [37].

Initial treatment with echinocandin and anidulafungin have been found to be associated with better candidemia clearance and less likely treatment failure [20,38], but most of them were adult studies, and the results remain conflicting [20,38,39]. However, other studies did not support these results and found that the key to successfully treating candidemia would be adequate source control and appropriate antifungal agent administered in time [39,40]. In our cohort, we found that antifungal regimens did not have a significant influence on the treatment outcomes, whereas the final outcomes depended on the presences of chronic comorbidities and septic shock at the onset and whether the attending physician had early catheter removal. In our institute, echinocandins were rarely used as the initial treatment, and azoles remained the first-line antifungal agents because most *Candida* isolates were susceptible to fluconazole or amphotericin B. It is well known that the formation of biofilm will cause persistent candidemia [41], even though appropriate antifungal agents have been given. Therefore, we suggested that echinocandins or Amphotericin B should be considered as the therapeutic regimen, as the updated published European guidelines recommend [42], because these antifungal agents are effective in vitro against biofilms.

Different *Candida* species and the presences of chronic comorbidities could potentially affect the therapeutic response and the outcomes, as other studies have concluded [5,43]. Previous studies found that non-albicans Candida species were more likely to be antifungal resistant and potentially associated with a worse outcome [43,44,45], but this is not proven in this study. Among all the neonates with chronic comorbidities, the neonates with neurological sequelae and severe bronchopulmonary dysplasia with secondary pulmonary hypertension were most likely to have the worse outcomes, although the sample size in our cohort was inadequate to have significant statistical power. We found that these patients finally died of cardiopulmonary failure or another episode of clinical sepsis even though candidemia clearance was achieved. Recurrent sepsis in neonates has been documented to account for the high mortality rate in the NICU [46].

Only 14.5% of the Candida isolates in our cohort were azole resistant, which was relatively higher than other studies. However, there is an increasing trend of non-albicans candidemia in our cohort, especially after progressive antifungal prophylaxis since 2016 in our NICUs. Therefore, the azole-resistant rate of the Candida isolates has also been relatively higher since uncommon Candida species became more prevalent, and most of them were unsusceptible to fluconazole. Previous studies have also documented the increasing azole-resistant rate after routine antifungal prophylaxis [47,48]. We suggested that the policy of antifungal prophylaxis, especially among neonates of extremely low birth weight, should be judged based on more benefits than harm before clinical applications [14,49].

Some limitations need to be addressed in this study. Because of the single center and the retrospective study design, the conclusion of this study may be less applicable to other countries, where the epidemiological characteristics may be different. The case number of this cohort is relatively inadequate, and the study period extended for nearly two decades, during which time different therapeutic strategies may have affected the outcomes. It is inevitable that the inadequate cases may preclude more subgroup analyses, such as specific non-albicans Candida species or antifungal-resistant Candida species as the pathogens. Additionally, serial follow-up blood culture was not performed systemically in most candidemia episodes, and the treatment strategies may be affected by the bias of a higher severity of illness.

In conclusion, we identified that delayed catheter removal, neonates with multiple chronic comorbidities, and septic shock were the most important predictors of final in-hospital mortality. The candidemia-attributable mortality in neonates remains high, perhaps due to a higher severity of illness and the delayed use of effective antifungal agents. The treatment outcomes have not changed or improved over the past decade, although new antifungal agents, such as caspofungin, have been launched in recent years. Therefore, prompt early catheter removal and aggressive therapeutic strategies for neonates with candidemia, especially for those with multiple chronic comorbidities or septic shock at the onset, will be very important to optimize the outcomes.

## Figures and Tables

**Figure 1 jof-08-00465-f001:**
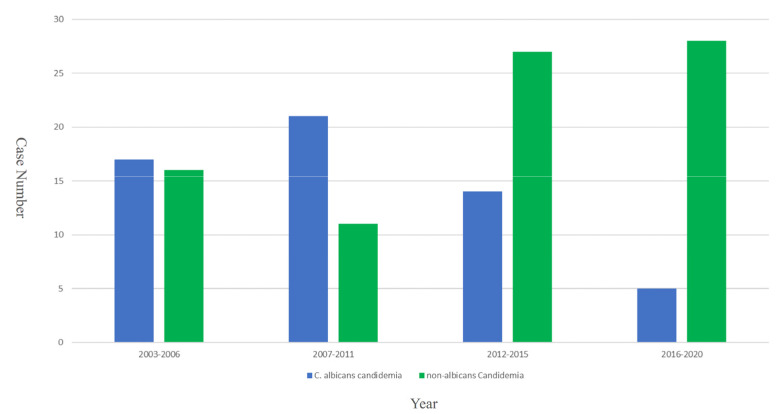
The increasing trend of non-albicans candidemia in the neonatal intensive care unit of Chang Gung Memorial Hospital, 2004–2020.

**Figure 2 jof-08-00465-f002:**
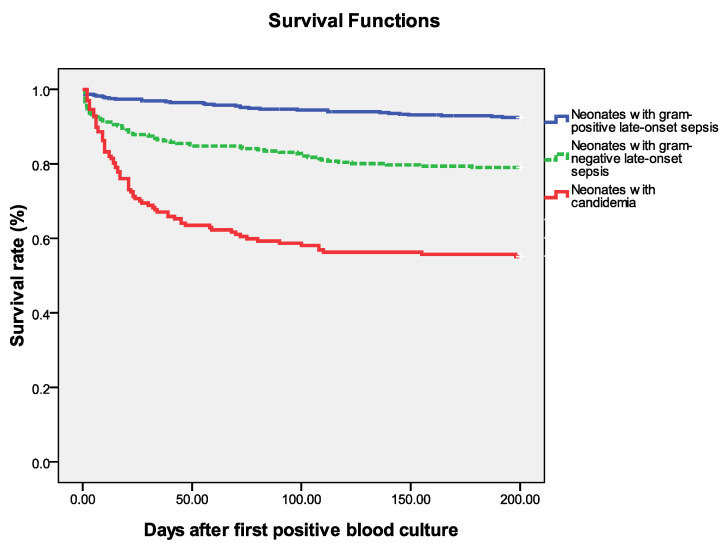
Survival following positive blood culture from 124 neonates with candidemia and 713 neonates with late-onset sepsis in the NICU. The Kaplan–Meier graph is stratified by Gram-positive, Gram-negative organisms, and candida species. The *y*-axis is the proportion surviving.

**Table 1 jof-08-00465-t001:** Patients’ demographics, microbiological characteristics, clinical features, and outcomes of 124 neonates with 139 episodes of candidemia in the neonatal intensive care unit of CGMH, 2003–2020.

Patients’ Characteristics (Total *n* = 124 Patients)	No. (%)	Microbiological Characteristics (Total *n* = 139 Episodes)	No. (%)
Patients’ demographics		Pathogens	
Birth body weight (g), median (IQR)	1104.0 (762.0–2055)	* Candida albicans*	57 (41.0)
Gestational age (weeks), median (IQR)	29.0 (26.0–35.0)	* Candida parapsilosis*	44 (31.7)
Gender (male/female)	75 (60.5)/49 (39.5)	* Candida tropicalis*	5 (3.6)
NSD/Cesarean section	60 (48.4)/64 (51.6)	* Candida glabrata*	11 (7.9)
Inborn/outborn	93 (75.0)/31 (25.0)	* Candida guilliermondii*	12 (8.6)
5 min Apgar score ≤ 7, *n* (%)	52 (41.9)	Other *Candida* spp.	10 (7.2)
Perinatal asphyxia, *n* (%)	8 (6.5)	Source of candidemia *	
Respiratory distress syndrome (≥Gr II), *n* (%)	64 (51.6)	Primary bloodstream infection (BSI)	92 (66.2)
Intraventricular hemorrhage (≥Stage II), *n* (%)	28 (22.6)	Catheter-related BSI	37 (26.6)
Day of life at onset of candidemia (day), median (IQR)	26.0 (15.0–61.0)	Abdominal	14 (10.1)
Underlying Chronic Comorbidities, *n* (%)		Urological	4 (2.9)
Neurological sequelae	31 (25.0)	Pulmonary	7 (5.0)
Bronchopulmonary dysplasia	70 (56.5)	Meningitis	3 (2.2)
Complicated cardiovascular diseases	6 (4.8)	Clinical presentation	
Symptomatic patent ductus arteriosus	27 (21.8)	Sepsis	112 (80.1)
Gastrointestinal sequelae	31 (25.0)	Severe sepsis	77 (55.4)
Renal disorders	8 (6.4)	Septic shock	55 (39.6)
Congenital anomalies	18 (14.5)	Progressive and deteriorated ^¶^	41 (29.5)
Presences of any chronic comorbidities	103 (83.1)	Disseminated candidiasis ^$^	12 (8.6)
Presences of more than one comorbidities	47 (37.9)	Duration of candidemia, median (range)	4.0 (1.0–42.0)
Case years		Predisposing risk factors ^#^	
2003–2006	33 (23.7)	Receipt of systemic antibiotics ^&^	129 (92.8)
2007–2011	32 (23.0)	Previous azole exposure ^&^	22 (17.1)
2012–2015	41 (29.5)	Prior bacteremia ^&^	54 (38.8)
2016–2020	33 (23.7)	Presence of CVC	133 (95.7)
30-day, all-cause mortality (from the first episode)	40 (32.3)	Receipt of parenteral nutrition	130 (93.5)
Within 72 h without receiving antifungal therapy	5 (4.0)	Receipt of immunosuppressants	4 (2.9)
Early mortality (>3 days, ≤7 days)	9 (7.3)	Artificial device other than CVC	42 (30.2)
Late mortality (8–30 days)	26 (21.0)	Prior surgery ^&^	45 (32.4)
Candidemia-attributable mortality	30 (24.2)	Neutropenia (ANC< 0.5×10^3^/µL)	11 (7.9)
Overall final in-hospital mortality	53 (42.7)	Persistent BSI ≥ 72 h after therapy	52 (37.4)
		Clinical treatment failure	47 (33.8)

* Defined as the first sterile site where the fungal culture was positive, and sometimes multiple sources were identified in one candidemia episode. ^¶^ Defined as candidemia episodes with more disseminated candidiasis and/or progressive multi-organ failure even after effective antifungal agents. ^#^ Indicated the presence of underlying condition or risk factor at onset of *Candida* BSI, and most episodes occurred in patients with >1 underlying condition or risk factor. ^&^ Within one month prior onset of candidemia. ^$^ Indicated positive *Candida* isolates recovered from more than two sterile sites, in addition to primary bloodstream infection. BSI: bloodstream infection; ANC: absolute neutrophil count; CVC: central venous catheter; IQR: interquartile range; SD: standard deviation.

**Table 2 jof-08-00465-t002:** Minimum inhibitory concentration distributions of six antifungal agents to all Candida isolates that caused neonatal candidemia in the NICU of Chang Gung Memorial Hospital, 2003–2020.

Pathogens/Antifungals	No. of Isolates with MIC (mg/L) of:											MIC (mg/L)		
	0.008	0.015	0.03	0.06	0.12	0.25	0.5	1.0	2.0	4.0	≥8.0	GM	MIC_50_	MIC_90_
*Candida albicans* (total *n* = 57 tested)														
Amphotericin B						4	50	3				0.494	0.5	0.5
Fluconazole					2	14	32	7			2	0.435	0.5	1
Voriconazole	44	10	1						2			0.011	≤0.008	0.015
Micafungin	34	13	8	1	1							0.012	≤0.008	0.03
Caspofungin		2	9	40	5	1						0.056	0.06	0.12
Anidulafungin		14	12	24	7							0.040	0.06	0.12
*Candida parapsilosis* (total *n* = 44 tested)														
Amphotericin B						4	27	13				0.576	0.5	1.0
Fluconazole					1	3	19	17	4			0.685	1.0	1.0
Voriconazole	10	21	10	3								0.017	0.015	0.03
Micafungin			1				7	25	10	1		0.999	1.0	2.0
Caspofungin					1	3	27	11	2			0.585	0.5	1.0
Anidulafungin						1	6	28	8	1		1.032	1.0	2.0
*Candida glabrata* (total *n* = 11 tested)														
Amphotericin B							4	7				0.777	1.0	1.0
Fluconazole											11	8.000	≥8.0	≥8.0
Voriconazole						2	6	2	1			0.567	0.5	1.0
Micafungin	2	8								1		0.022	0.015	0.015
Caspofungin			1	2	7						1	0.137	0.12	0.12
Anidulafungin			6	4					1			0.057	0.03	0.06
*Candida tropicalis* (total *n* = 5 tested)														
Amphotericin B							1	4				0.871	1	1
Fluconazole								1	2	1	1	2.639	2	8
Voriconazole					2	2	1					0.214	0.25	0.5
Micafungin		1	3	1								0.030	0.03	0.06
Caspofungin			1	2	1	1						0.080	0.06	0.25
Anidulafungin			1	1	2	1						0.092	0.12	0.25
Other *Candida* spp. (total *n* = 22 tested)														
Amphotericin B					2	9	7	3	1			0.387	0.5	1.0
Fluconazole					1	1	1	3	5	6	5	2.264	2	8
Voriconazole	3	1	3	7	5	2					1	0.065	0.06	0.25
Micafungin		3	1	2	1	5	8	2				0.192	0.25	1
Caspofungin			2	3	3	9	4				1	0.204	0.25	0.5
Anidulafungin		2	1	2	2	3	3	7	2			0.298	0.5	1

MIC: minimum inhibitory concentration. MIC_50_ and MIC_90_: MIC required to inhibit 50% and 90% of the isolates, respectively. GM: geometric mean.

**Table 3 jof-08-00465-t003:** Therapeutic strategies of neonatal candidemia and comparisons of the *C. albicans* candidemia versus non-albicans candidemia.

Variable	Overall (Total *n* = 139)	*C. albicans* Candidemia (Total *n* = 57)	Non-Albicans Candidemia (Total *n* = 82)	*p* Value ^¶^
Final treatment regimens				0.870
Fluconazole/Voriconazole	42 (30.2)	16 (28.1)	26 (31.7)	
Amphotericin B	57 (41.0)	25 (43.9)	32 (39.0)	
Echinocandin-based regimen	27 (19.4)	10 (17.5)	17 (20.7)	
Combination antifungal treatment	7 (5.0)	3 (5.3)	4 (4.9)	
No treatment	5 (3.6)	3 (5.3)	2 (2.4)	
Modification of antifungal agents	55 (39.6)	24 (42.1)	31 (37.8)	0.478
Duration of antifungal treatment (d), median (IQR)	17.0 (14.0–24.0)	16.0 (14.0–22.0)	18.0 (14.0–24.0)	0.561
Early removal of central venous catheter *	50 (36.0)	20 (35.1)	30 (36.6)	0.856
Treatment outcomes				
Responsiveness after effective antifungals ^&^				
Within 72 h	48 (34.5)	23 (40.4)	25 (30.5)	0.209
4–7 days	20 (14.4)	8 (14.0)	12 (14.6)	0.988
More than 7 days	24 (17.3)	8 (14.0)	16 (19.5)	0.623
Treatment failure	47 (33.8)	18 (31.6)	29 (35.4)	0.356
Duration of candidemia (day), median (IQR)	4.0 (2.0–8.0)	4.0 (1.0–7.0)	4.0 (2.0–9.0)	0.233

* Within the 3 days after onset of candidemia. **^¶^**
*p* values were the comparison between neonatal episodes and non-neonatal episodes. ^&^ Responsiveness to antifungal agents was defined according to the consensus criteria of the Mycoses Study Group and the European Organization for Research and Treatment of Cancer [26].

**Table 4 jof-08-00465-t004:** Univariate and multivariate logistic regression analysis for independent risk factors of final in-hospital mortality in neonates with candidemia.

Variables	Univariate Analyses			Multivariate Regression Analysis *		
	Odds Ratio	95% CI	*p* Value ^#^	Odds Ratio	95% CI	*p* Value ^#^
Gestational age						
≤27 weeks	1.79	0.75–4.26	0.071			
28–32 weeks	0.39	0.14–1.09	0.190			
≥33 weeks	1	(reference)				
Underlying chronic comorbidities						
No	1	(reference)		1	(reference)	
One	2.84	0.74–10.9	0.128	3.63	0.77–17.01	0.102
More than one chronic comorbidity	12.8	3.26–50.25	<0.001	8.71	1.82–41.81	0.007
Septic shock	11.55	4.87–27.40	<0.001	7.88	2.83–21.93	<0.001
Delayed CVC removal (>72 h)	6.48	2.67–15.69	<0.001	5.54	1.93–15.86	0.001
Subsequent bacteremia	1.53	0.75–3.13	0.244			
Breakthrough candidemia	3.17	1.10–9.12	0.032	0.97	0.24–4.06	0.974
Delayed effective antifungal agents (>48 h)	1.38	0.65–2.91	0.400			
Final antifungal therapy						
Fluconazole/Voriconazole	1	(reference)				
Amphotericin B	0.87	0.37–2.07	0.757			
Echinocandin-based regimen	1.75	0.62–4.95	0.295			
Combination regimens	3.20	0.52–19.67	0.209			
Pathogens						
* Candida albicans*	1	(reference)				
* Non-albicans Candida* spp.	0.93	0.46–1.92	0.857			
Uncommon *Candida* spp.	1.40	0.69–2.85	0.354			
Case periods						
2003–2006	1.63	0.60–4.41	0.335			
2007–2011	1.85	0.65–5.24	0.246			
2012–2015	0.86	0.30–2.50	0.787			
2016–2020	1	(reference)				

CI: confidence interval; CVC: central venous catheter. * For patients with more than two episodes of *Candida* bloodstream infection, data from the first episode of *Candida* bloodstream infection were enrolled into the model of multivariate analysis for predictors of final in-hospital mortality. ^#^ Hosmer–Lemeshow *p =* 0.649 and 0.427 for fungemia-attributable mortality and in-hospital mortality, respectively.

## Data Availability

The datasets used/or analyzed during the current study are available from the corresponding author on reasonable request.

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
