# Peer review of "Clinical and Microbiological Characteristics of Neonates with Candidemia and Impacts of Therapeutic Strategies on the Outcomes"

_jof, 2022, doi:10.3390/jof8050465_

Round 1

Reviewer 1 Report

Chen et al. performed a single-center study in order to investigate the clinical characteristics of Candida bloodstream infections in neonates and the impact of therapeutic strategies on outcomes. Despite its monocentric approach, this is a well-written study.

Minor comments:

  1. were there any differences in antifungal susceptibility during the prolong study period?

Reviewer 2 Report

The manuscript entitled: “Clinical and microbiological characteristics of neonates with candidemia and impacts of therapeutic strategies on the outcomes” by Yu-Ning Chen et al., is a well written study concerning the candidemia in neonates.

It is an original article of great importance in its field that could be accepted as it is..

It is a well written retrospective study for 18-year period (2003-2021) in a special population, those of “neonates”. The manuscript is very interesting; the methodology used is up to date and it gives information about a special and very sensitive population for a long period of time.

Recommendation: accept submission

Best regards

Author Response

Dear reviewer:

Thank you for your appreciated comments. I appreciate your review, thank you.

Best regard,

Tsai Ming Horng
